# Myoinositol and Metformin in the Prevention of Gestational Diabetes in High-Risk Patients: A Narrative Review

**DOI:** 10.3390/jcm13185387

**Published:** 2024-09-12

**Authors:** Paola Romeo, Rosario D’Anna, Francesco Corrado

**Affiliations:** Obstetrics and Gynecology Unit, Department of Human Pathology, University of Messina (Italy), Via Consolare Valeria, 1, 98125 Messina, Italy; paolaromeo135@gmail.com (P.R.);

**Keywords:** myoinositol, metformin, insulin resistance, prevention of gestational diabetes

## Abstract

Our hypothesis is that myoinositol and metformin in pregnant women with high-risk factors for glucose intolerance would reduce insulin resistance and consequently lower the incidence of gestational diabetes mellitus (GDM), a metabolic disorder of pregnancy characterized by maternal hyperglycemia due to deficient response to physiological insulin resistance, which may have a negative impact on perinatal outcome and long-term sequelae. In recent years, this pathology has become increasingly important given the global obesity epidemic and the delay in becoming pregnant, especially in industrialized countries. For this reason, the attempt to prevent, rather than cure, gestational diabetes is particularly important. In addition to lifestyle changes (especially diet and doing more exercise), myoinositol and metformin are the most promising factors at the moment, although not all RCTs published so far agree on their real effectiveness. A review of the articles published so far allows us to assume, albeit with some distinctions, that they can play a positive role.

## 1. Introduction

Gestational diabetes mellitus (GDM) is a form of carbohydrate intolerance that is first detected during pregnancy, but after the first trimester of gestation [1]. In most cases, hyperglycemia results from an impaired glucose tolerance due to chronic insulin resistance. The prevalence of GDM is rising annually, driven by the global obesity epidemic, making it a significant health concern, with serious implications for both maternal and neonatal outcomes. GDM underestimation during pregnancy [2] leads to an increased risk of gestational hypertension, preterm birth, cesarean section, fetal macrosomia, large-for-gestational-age (LGA) babies, shoulder dystocia, neonatal hypoglycemia, neonatal respiratory distress, neonatal jaundice requiring phototherapy, and babies hospitalization in neonatal intensive care unit (NICU). Furthermore, GDM has long-term consequences, including an elevated risk of cardiovascular disease, with a twofold higher risk (RR 1.98 [95% CI 1.57–2.50]) of cardiovascular events postpartum, compared with their peers [3], and an elevated risk of type 2 diabetes (T2DM) for mothers with a history of GDM, who appear to have a nearly 10-fold higher risk (RR 9.51, [95% CI 7.14–12.67]) of developing T2DM compared to those with a normoglycemic pregnancy [4]. Additionally, there is a trend towards obesity and a high rate of overweight during follow-up [5], as well as an increased risk of cardiovascular diseases and diabetes in children [6]. While diagnosing and treating GDM with lifestyle interventions, metformin, and/or insulin have been shown to reduce maternal and perinatal morbidity [7,8], an effective prevention strategy remains elusive. Indeed, categories at greater risk of GDM have been identified: family history of diabetes, overweight or obesity, advanced maternal age, previous GDM, previous macrosomia, polycystic ovary syndrome (PCOS), belonging to high-risk ethnic groups, having fasting blood glucose values higher than average but not high enough to diagnose diabetes (<126 mg/dL). Despite identifying these risk factors, the conflicting results on the effectiveness of dietary supplements or drugs, such as myoinositol and metformin, make it unclear whether their intake could reduce the prevalence of GDM. In this narrative review, we aim to revisit the extensive research conducted, unfortunately with mixed results, on myoinositol and metformin as potential prophylactic agents for GDM, particularly for patients at high risk.

## 2. Myoinositol for GDM Prevention

Inositol is a cyclic polyol with nine stereo-isomeric forms, which is linked to phospholipids in the membranes of all living cells. The most abundant isomer is myoinositol (MI), which is also found in various foods. Fresh fruits and vegetables contain the greatest amounts of MI, particularly beans, peas, cantaloupe melons, and citrus fruits. MI is produced by the human body from D-glucose or is obtained from the breakdown of inositol-containing membrane phospholipids, and it can also be taken up from the extracellular fluid via MI transporters [9]. The kidneys are the primary regulators of MI metabolism, each kidney producing about 2 g daily, which exceeds the average dietary intake of about 1 g. MI is catabolized only in the kidneys. A specific epimerase present in tissues converts MI to another isomer, D-chiro-inositol (DCI). Both MI and DCI are involved in glucose metabolism as components of glycosyl-phosphatidylinositol (GPI) and of inositol phosphoglycans (IPGs), which are second messengers of insulin action in the GPI/IPG pathway [10]. Furthermore, it seems that MI-PG and DCI-PG mediate different actions of insulin: MI-PG mediates cellular glucose uptake by activating Glucose Transporter Type 4 (GLUT-4) translocation to the plasma membrane, allowing glucose to enter the cells. In contrast, DCI-PG is involved in glycogen synthesis. It was demonstrated that in individuals with diabetes mellitus (DM), an imbalance between MI and DCI is present, with a decreased urinary excretion of DCI and an increased urinary excretion of MI [11]. Experimental studies have shown that in hyperglycemic conditions, such as in DM, the glucose-induced MI uptake inhibition results from a competition between MI and glucose for MI transporters, as both exhibit structural similarities [12]. Depletion of intracellular MI and, consequently, decreased production of DCI from MI reduces the availability of these two substances, limiting their incorporation into IPGs, affecting insulin signaling [11]. A recent experimental study carried out on 24 placentas from GDM women, with the aim to evaluate the glucose effects on placental expression of inositol enzymes and transporters, evidenced a reduced concentration of placental inositol of about 17% in GDM pregnancies. These data were associated with lower protein levels of the myoinositol synthesis enzyme and inositol transporters and were inversely correlated with fasting glycemia [13]. All these considerations suggest that MI and/or DCI supplementation could be therapeutic for pregnant women at risk of GDM to restore depleted tissue levels of these substances. [14]. DCI administered to obese PCOS women for at least 6 months improved ovarian function [15]. These preliminary results were not subsequently confirmed, but the treatment of infertile PCOS women was continued with the other inositol isomer, myoinositol. In a retrospective study [16] on women with PCOS, two groups of 98 infertile PCOS women were treated with two different insulin-sensitizing substances, myoinositol or metformin, to normalize their cycles and ovulation. After a positive pregnancy test, metformin was discontinued, and these women served as the control group, while the MI group continued the treatment throughout pregnancy. All these pregnant women performed an OGTT at 24–28 weeks of gestation, highlighting a significant difference in the rate of GDM diagnosis between the two groups: 17.4% in the MI group vs. 54% in the control group [16]. Even retrospectively, this study demonstrated for the first time that MI supplementation from the first trimester may reduce the GDM rate. Subsequently, three randomized controlled trials were conducted, involving women with three different GDM risk factors: family history of type 2 diabetes [17], obesity [18], and overweight [19]. The participants received randomly either MI (2 g plus 200 mcg folic acid twice daily) or a placebo (200 mcg folic acid twice daily), from the end of the first trimester until delivery. In all trials, the treated group showed a statistically significant reduction in GDM rates. Although all these trials were randomized and controlled, the decision imposed by the Ethics Committee to authorize them open-label rather than double-blind could partly cast doubt on the conclusions reached. A secondary analysis of the combined data from these trials, involving a total of 660 pregnant women, confirmed a 66% reduction in GDM rates in the treated group [20]. Furthermore, analyzing the group of treated women (n.32, 11%) in which myoinositol failed to prevent GDM, it is noteworthy that this group of women with GDM had, in the first trimester, a statistically (*p* = 0.01) significant higher insulin resistance (HOMA = 2.97 ± 1.7) compared to the placebo GDM group (HOMA = 2.30 ± 2.2). Other significant differences were evidenced for the weight gain at OGTT (kg. 7.2 ± 4.1 vs. 5.5 ± 4.3, *p* = 0.02), which was higher in the treated group and for the gestational age at delivery (*p* = 0.002), which was anticipated. Thus, the failure of myoinositol’s preventing effect on high-risk pregnancies for GDM might be due to a disadvantaged metabolic condition at the beginning of pregnancy, such as a high level of insulin resistance. Additionally, univariate and multivariate regression analyses indicated that MI supplementation not only reduced GDM rates but also pre-term birth rates, with borderline statistical significance for large-for-gestational-age (LGA) infants and gestational hypertension. Similar results were found in other studies [21,22], though the efficacy of MI seemed to be dose-effective. For instance, another RCT study, using a daily combination of 1100 mg MI, 27.6 mg of DCI, and 400 mcg of folic acid, showed no significant difference in GDM rates compared to the placebo group [23]. The cited intervention studies are summarized in Table 1. In a very recent meta-analysis [24], including 1795 patients, reduced incidence of GDM occurrence by myoinositol supplementation was confirmed (RR = 0.42, CI: 0.26–0.67). Furthermore, a reduced risk of insulin need (RR = 0.29, CI: 0.13–0.68), preeclampsia, or gestational hypertension (RR = 0.38, CI: 0.2–0.71), preterm birth (RR = 0.44, CI: 0.22–0.88), and neonatal hypoglycemia (RR = 0.12, CI: 0.03–0.55) was also shown. Another meta-analysis [25] including seven RCTs with 1319 pregnant women at high risk of GDM evidenced that MI supplementation significantly reduced the GDM rate by 60% (OR 0.40) versus the control group. Furthermore, MI improved the glycemic values of the OGTT and reduced the risks of pregnancy-induced hypertension (OR 0.37), pre-term birth (OR 0.35), and neonatal hypoglycemia (OR 0.10).

## 3. Metformin for GDM Prevention

Metformin is an antidiabetic agent, approved by the U.S. Food and Drug administration (FDA) in 1995 for treating type 2 diabetes. It belongs to the biguanide family, and its mechanism of action is linked to the activation of an AMP kinase. This activation lowers blood glucose levels by inhibiting hepatic gluconeogenesis and reducing glucose absorption at the intestinal level [26,27]. Additionally, metformin may alter the composition and functions of the gastrointestinal microbiome [28], which is fundamental for activating the gut–brain–liver feedback system that regulates nutrient sensing and hepatic glucose production. Consequently, metformin reduces insulin sensitivity without causing weight gain or hypoglycemia. Most common side effects are gastrointestinal, such as diarrhea, nausea, and stomach pain. Although its use for treating gestational diabetes was hypothesized early on, when initial studies, conducted in the late 1970s, proposed metformin for the treatment of preexisting type 2 diabetes or gestational diabetes when dietary treatment had failed [29,30], the lack of data on health impacts on pregnant women and the fetuses initially prevented its clinical use. This changed in 2008, when Rowan JA et al. [31] published an RCT study in the New England Journal of Medicine involving 751 pregnant women: 373 received metformin, and 378 received insulin, with supplemental insulin needed for 168 (46.3%) pregnant women in the metformin group to achieve glycemic targets. The primary outcome was a composite of neonatal hypoglycemia, respiratory distress, need for phototherapy, birth trauma, 5 min Apgar score less than 7, or prematurity. The study concluded that metformin, alone or in combination with insulin, is an effective and safe option for women with gestational diabetes who require something else, to meet their treatment goals, whether with metformin alone or with supplementary insulin. Since then, numerous studies have confirmed metformin’s effectiveness and safety [32] for the treatment of gestational diabetes mellitus (GDM), particularly for high-risk patients such as overweight or obese women, highlighting its better compliance compared to insulin. In consideration of this, in 2015, the National Institute for Health and Care Excellence of the United Kingdom (NICE) [33] proposed the use of metformin as first-line therapy when lifestyle modifications (diet and doing more exercise) alone do not meet glycemic targets after a GDM diagnosis. Moreover, a randomized study on 2190 patients (350 with a past history of GDM and 1416 with a previous live birth but no history of GDM) [34] demonstrated that postpartum treatment with metformin effectively delayed or prevented diabetes mellitus type II, reducing the incidence of diabetes mellitus later in life by approximately 50% (*p* = 0.006) in women who reported a history of GDM, compared with the placebo group. The study was a multicenter, National Institutes of Health-sponsored trial, carried out at 27 centers, including academic and Indian Health Service sites. While metformin’s effectiveness in treating GDM is now widely accepted, its use as a prophylactic measure in at-risk cases remains uncertain. Initial studies were conducted on patients with polycystic ovary syndrome (PCOS), a dysmetabolic condition characterized by insulin resistance at the level of target tissues (adipose tissue, muscles, and liver). This endocrine disorders affect approximately 10% of all women during the reproductive age. In most cases, these are patients who conceived while taking metformin, and the studies have often compared pregnant women who continued this therapy throughout pregnancy with those who did not. Early data were conflicting: Fougner [35] and Vanky [36] found no effect on glucose homeostasis and reduced incidence of GDM (17.6% in the metformin group and 16.9% in the placebo group *p* = 0.87). Few years later (2019) a randomized, placebo-controlled, double-blind, multicenter trial conducted at 14 hospitals in Norway, Sweden, and Iceland [37] randomly assigned the 487 pregnant patients to metformin (*n* = 244) or placebo (*n* = 243). The participants assigned to receive metformin assumed 500 mg twice daily during the first week, which increased to 1000 mg twice daily from week 2 until delivery, while the others assumed placebo tablets that were identical to metformin tablets. They found no significant differences for gestational diabetes incidence (60 [25%] in the metformin group vs. 57 [24%] in the placebo group; OR 1.09, 95% CI 0.69–1.66; *p* = 0.75). At the same time, other authors, including Begum [38] and Khattab [39], concluded that the metformin used throughout pregnancy is associated with a significant reduction in the prevalence of GDM. Specifically, Begum [38] reported an OR for GDM of 12 in PCOS women without metformin versus with metformin (95% confidence interval: 6.20–18.08), with a ninefold reduction (30% vs. 3.4%) in gestational diabetes, while Khattab [39] reported a statistically significant reduction in the incidence of GDM in favor of the metformin group (OR: 0.17, 95% CI: 0.07–0.37), as well as a statistically significant reduction in the incidence of preeclampsia (OR: 0.35, 95% CI: 0.13–0.94). Lastly, Ainuddin conducted a comparative cohort study in 2015 [40] on patients affected by polycystic ovarian syndrome, who became pregnant. The patients were divided into two groups, one of which involved taking metformin. The study reported that 10% of cases in the metformin group developed GDM, while 34.3% of cases in the no-metformin group developed GDM. In other words, patients not receiving metformin were 4.7 times more likely to have GDM (OR: 4.7, *p* = 0.01) compared to those who received it. The only difference between the two groups seems to be ethnicity, with Caucasian patients in the former and Afro-Asian patients in the latter. This, in light of future studies, could be something that needs to be explored further and could justify the different outcomes. Other studies were conducted on the use of metformin in pregnant patients with various risk factors, such as obesity [41]. In a cohort of 51 pregnant women at high risk to develop GDM [42], metformin had no significant effect on preventing GDM. Gestational diabetes mellitus was diagnosed in 15.9% of the obese patients randomly allocated to the metformin group and 19.5% of those in the control group, with an absolute risk reduction which was not statistically significant (*p* = 0.63). Lastly [43] administration of metformin in patients with pregestational insulin resistance (PIR) was not associated with a decrease in the incidence of GDM, as compared to placebo (37.5% vs. 25.4%, respectively; *p* = 0.2). Moreover metformin administration was associated with a significant increase in drug intolerance, as compared to placebo (14.3% vs. 1.8, respectively; *p* = 0.02). In conclusion, taken as a whole, although the rationale suggested otherwise, they were not able to confirm a significant prophylactic effect of Metformin on the incidence of GDM. A 2020 overview of Cochrane Reviews by Rebecca J Griffin [44] about interventions (diet, exercise, a combination of diet and exercise, dietary supplements and pharmaceuticals) to prevent GDM in high-risk categories hypothesized a modest protective effect for obese women, but this finding was not statistically significant. This conclusion was primarily based on two RCTs [45,46] conducted on obese women, which failed to demonstrate a real benefit (RR 0.85, [95% CI 0.61–1.19], *p* = NS). The first study, by Syngelaki et al. [45], a double-blind, placebo-controlled trial, randomized over 450 obese pregnant women (BMI > 35) to take a dose of 3 g of metformin per day starting from the 12th to 18th week of gestation. No significant differences were reported between groups regarding the incidence of GDM (25/202 vs. 22/195 *p* = 0.7) or the percentage of LGA neonates (34/202 vs. 30/195 *p* = 0.7). However, there was a significantly reduced maternal median weight gain with metformin administration (4.6 kg vs. 6.3 kg *p* = 0.001). The second study by Chiswick C. et al. [46], a double-blind, placebo-controlled trial, included 449 pregnant obese patients (BMI ≥ 30) randomized to receive oral metformin 500 mg (increasing to a maximum of 2500 mg) or placebo, with no difference regarding the incidence of GDM between groups (23.5% vs. 18.3% *p* = 0.27). In the latter trial, unfortunately, a low percentage (around 15%) of women eligible for the study was recorded.

All the cited intervention studies are reported in Table 2.

However, the final word on this topic has likely not yet been said. A recent review (2024) by a joint ADA/EASD panel [47], which included a further review of published RCTs, stated that metformin might reduce the risk of developing GDM by 34% in certain risk categories (RR 0.66; [95% CI 0.47–0.93]). However, further subgroup analysis was not undertaken due to an insufficient number of studies on each group of intervention characteristics. Indeed, another recent (2024) meta-analysis [48] highlighted an interesting consideration: not all interventions work equally for all participants; therefore, it is essential to consider specific characteristics such as ethnicity, obesity, history of GDM or PCOS, and metformin dosage (ranging among the studies from 500 to 3000 mg/day) to clarify the specific effectiveness of the intervention. Lastly, it is also important to note that this meta-analysis may have been affected by bias due to the inclusion of non-RCT studies.

## 4. Conclusions

Myoinositol and Metformin, as well as lifestyle interventions, administered during the first half of pregnancy, appear to play a positive role in the prevention of GDM. However, many different interacting parameters have often given conflicting conclusions on this topic. Ethnicity, the different risk factors considered, drug dosages, as well as a small sample size or a reduced adherence rate to the study, sometimes found, are probably the factors that most influence the different conclusions of the Authors, and prevented them from finding a shared protocol in the prevention of GDM in pregnant women with risk factors. Therefore, following these observations emerging from the analysis of the studies reported in the text, a multicenter effort will be necessary for both drugs to find the necessary sample size to be able to separate characteristics such as ethnicity, different risk factors, and dosages within the groups. Using a double-blind method, compared to open-label approach, if authorized by the Ethics Committees, would make the final evaluation more objective.

## Figures and Tables

**Table 1 jcm-13-05387-t001:** Intervention studies on myoinositol reported (cited in the text).

Author/Year	Study	Dosage	Start Treatment	Ethnicity	Risk Factor	Significance
D’Anna 2012 [16]	Cohort study	4 g Myo + 400 μg folic acid	Starting pregnancy	Caucasian	PCOS	OR 2.4 **; (95% CI 1.3–4.4)
D’Anna 2013 [17]	RCT	4 g Myo + 400 μg folic acid	From weeks 12–13	Caucasian	Family history of TDM2	OR 0.35 *; (95% CI 0.13–0.96)
D’Anna 2015 [18]	RCT	4 g Myo + 400 μg folic acid	From weeks 12–13	Mixed	BMI > 30	OR 0.34 *; (95% CI 0.17–0.68)
Santamaria 2016 [19]	RCT	4 g Myo + 400 μg folic acid	From weeks 12–13	Caucasian	Overweight (BMI 25–29.9)	OR 0.33 *; (95% CI 0.15–0.70)
Matarelli 2013 [21]	RCT	4 g Myo + 400 μg folic acid	Until 16 weeks	Caucasian	BMI < 30 and fasting gluc. >100 mg and <126 mg/dL	*p* = 0.001
Farren 2017 [23]	RCT	1.1 g MI + 27.6 mg DCI + 400 μg folic acid	From weeks 10–16	Mixed	Family history of TDM2	*p* = 0.34
Vitale 2021 [22]	RCT	4 g Myo + 400 μg folic acid	From weeks 12–13	Caucasian	Overweight (BMI 25–29.9)	OR 3.74 **; (95% CI 1.67–8.39) *p* = 0.001

Myoinositol vs. controls *. Controls vs. myoinositol **.

**Table 2 jcm-13-05387-t002:** Intervention studies on metformin reported (cited in the text).

Author/Year	Study	Dosage	Start Treat	Ethnicity	Risk Factor	Significance
Fougner 2008 [35]	RCT	850 mg/twice/day	First Trimester	Caucasian	PCOS	*p* = NS
Vanky 2010 [36]	RCT	2000 mg/day	First trimester	Caucasian	PCOS	*p* = NS
Lǿwik 2019 [37]	RCT	1000 mg–2000 mg/twice/day	First trimester	Caucasian	PCOS	*p* = NS
Begum 2009 [38]	RCT	1500–2500 daily according BMI	First trimester	Bangladesh	PCOS	OR 12 * (95% CI 6.2–18.0)
Khattab 2011 [39]	Cohort study	1000–2000 mg/day	First trimester	Egyptian	PCOS	OR 0.17 ** (95% CI 0.07–0.37)
Ainnudin JA 2015 [40]	Cohort study	500–2500 mg/day	First trimester	Pakistan	PCOS	OR 4.7 * *p* = 0.01
Brink 2018 [41]	RCT	500–1000 mg twice/day	Second trimester	Caucasian	BMI > 30	*p* = 0.35
Sales 2018 [42]	RCT	1000 mg twice/day	Within 20 weeks	Brazil	BMI > 30	*p* = NS
Valdes 2018 [43]	RCT	850 mg/twice/day	Within 16 weeks	Chile	Pregestational insulin resistance	*p* = 0.2
Chiswick 2015 [46]	RCT	500–2500 mg/day	Within 16 weeks	UK (96% caucasian)	BMI > 30	*p* = 0.27
Syngelaki 2016 [45]	RCT	1000–3000 mg/day	Within 18 weeks	UK (70% caucasian)	BMI > 35	*p* = 0.74

Controls vs. metformin * and Metformin vs. controls **.

## Data Availability

No new data were created. All the data reported in the review are derived from the cited articles.

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
