# Peer review of "Myoinositol and Metformin in the Prevention of Gestational Diabetes in High-Risk Patients: A Narrative Review"

_jcm, 2024, doi:10.3390/jcm13185387_

Round 1

Reviewer 1 Report

Comments and Suggestions for Authors

Comments,

It is well known that GDM has many negative effects on the health of the mother and the unborn child in the short and long term, thus, although the present manuscript does not add new information about pharmacological intervention in GDM condition, a reorganization of the data shown in it could be helpful for future studies aimed at preventive interventions.

To this purpose, would the authors include a table reporting the intervention studies indicating, the dosage of myoinositol and  metformin, at what point in the pregnancy was the intervention performed, and its duration,  ethnicity and BMI ?

In addition a more detailed conclusion with hypothesis and suggestion for  future studies would make the manuscript interesting

Author Response

Answers to referees' questions

n.1

It is well known that GDM has many negative effects on the health of the mother and the unborn child in the short and long term, thus, although the present manuscript does not add new information about pharmacological intervention in GDM condition, a reorganization of the data shown in it could be helpful for future studies aimed at preventive interventions.

To this purpose, would the authors include a table reporting the intervention studies indicating, the dosage of myoinositol and  metformin, at what point in the pregnancy was the intervention performed, and its duration,  ethnicity and BMI ?

According to your suggestion 2 tables (one for each drug) of the intervention studies mentioned in the text have been inserted with the data of each study, the type of study, dosage, start of treatment (which was always continued until the end of the pregnancy), ethnicity, risk factor considered, statistical significance.

In addition a more detailed conclusion with hypothesis and suggestion for  future studies would make the manuscript interesting

According to your suggestion we have taken up in the conclusions the observations already made in the text and, based on the weaknesses of some, suggested what type of study could be useful to clarify the existing doubts about the actual effectiveness.

Reviewer 2 Report

Comments and Suggestions for Authors

In this manuscript, the authors present a narrative review on myoinositol and metformin in the prevention of gestational diabetes in high-risk patients. The authors conclude that myoinositol and metformin, as well as lifestyle interventions, taken during the first half of pregnancy, appear to play a positive role in the prevention of GDM. The topic is of great public health importance; I thank the authors for their submission. Suggestions:

1) The abstract, as written currently, looks like an introduction to GDM and does not provide sufficient details about key take-aways from this review. Please consider starting the abstract directly with a brief introduction (1 to 2 sentences) for the potential role of myoinositol and metformin in GDM and then present key take-aways from this review. 

2) In the introduction, it is mentioned that there are "mixed results" for "myoinositol and metformin as potential prophylactic agents for GDM"; but the conclusion says with a tone of certainty that myoinositol and metformin are effective... A narrative review is different from a systematic review because the authors get to pick and choose studies to include in their article which can lead to biased results. Therefore, going from "mixed results" to "certainly effective" is not appropriate for a narrative review. Please consider rewording the text 

3) Please consider defining the population under consideration- what is considered as "high risk" pregnancy in this review? was the review limited to first trimester of pregnancy? 

4) A table of included studies, summarizing findings from previous systematic reviews & meta-analyses would be helpful.

5) A figure illustrating potential underpinning mechanisms for the role of myoinositol & metformin in GDM prevention would be useful to readers.

6) Please consider discussing and citing the following systematic review & meta-analyses on myoinositiol:

Greff D, Váncsa S, Váradi A, et al. Myoinositols Prevent Gestational Diabetes Mellitus and Related Complications: A Systematic Review and Meta-Analysis of Randomized Controlled Trials. Nutrients. 2023 Sep 30;15(19):4224. doi: 10.3390/nu15194224. PMID: 37836508; PMCID: PMC10574514.

Thank you.

Author Response

Answers to referees' questions

n.2

In this manuscript, the authors present a narrative review on myoinositol and metformin in the prevention of gestational diabetes in high-risk patients. The authors conclude that myoinositol and metformin, as well as lifestyle interventions, taken during the first half of pregnancy, appear to play a positive role in the prevention of GDM. The topic is of great public health importance; I thank the authors for their submission. Suggestions:

1) The abstract, as written currently, looks like an introduction to GDM and does not provide sufficient details about key take-aways from this review. Please consider starting the abstract directly with a brief introduction (1 to 2 sentences) for the potential role of myoinositol and metformin in GDM and then present key take-aways from this review.

According to your suggestion we have included a sentence at the beginning of the abstract that immediately clarifies the objective of the study which is to verify whether metformin or myoinositol have any efficacy in preventing the onset of GDM; specifically we have added the following text “Our hypothesis is that myoinositol and metformin in pregnant women with high risk factor for glucose intolerance would reduce insulin resistance and consequently lower the incidence of gestational diabetes mellitus (GDM).”

2) In the introduction, it is mentioned that there are "mixed results" for "myoinositol and metformin as potential prophylactic agents for GDM"; but the conclusion says with a tone of certainty that myoinositol and metformin are effective... A narrative review is different from a systematic review because the authors get to pick and choose studies to include in their article which can lead to biased results. Therefore, going from "mixed results" to "certainly effective" is not appropriate for a narrative review. Please consider rewording the text

According to your suggestion we have already rewording the text replacing “certainly effective” with “appears to play a positive role in preventing GDM”. “Certainly effective” was present only in an old version and had been already removed. On the other hand, we have stressed several times in the text that the results are conflicting and that there is no certainty in this regard 

3) Please consider defining the population under consideration- what is considered as "high risk" pregnancy in this review? was the review limited to first trimester of pregnancy?

All the list of high risk are clearly specified in the text of the introduction  and we have specified the risk factor considered for each study in the tables  .

Not all the intervention studies are limited to the first trimester but some of them arrived until 20 weeks of gestation.

4) A table of included studies, summarizing findings from previous systematic reviews & meta-analyses would be helpful.

 According to your suggestion 2 tables (one for each drug) of the intervention studies mentioned in the text have been inserted with the data of each study, while we believe that including meta-analyses could lead to overlaps and misunderstandings

5) A figure illustrating potential underpinning mechanisms for the role of myoinositol & metformin in GDM prevention would be useful to readers.

According to your suggestions we have included a figure on the mechanism of action of myoinositol, while we believe that for metformin, that is part of biguanide family is not necessary

6) Please consider discussing and citing the following systematic review & meta-analyses on myoinositiol:

Greff D, Váncsa S, Váradi A, et al. Myoinositols Prevent Gestational Diabetes Mellitus and Related Complications: A Systematic Review and Meta-Analysis of Randomized Controlled Trials. Nutrients. 2023 Sep 30;15(19):4224. doi: 10.3390/nu15194224. PMID: 37836508; PMCID: PMC10574514.

According to your suggestion we included this study in the references and discussed it in the text.

Reviewer 3 Report

Comments and Suggestions for Authors

The reviewed manuscript is a narrative review discussing the potential roles of myoinositol and metformin in preventing gestational diabetes mellitus (GDM) in high-risk patients. The paper emphasizes the increasing prevalence of GDM due to rising global obesity rates and delayed pregnancies, particularly in industrialized countries. It explores the use of myoinositol and metformin as preventive measures, analyzing existing literature and studies to assess their efficacy.

While the review is comprehensive, it largely reiterates existing knowledge without offering new insights or critical analysis that could lead to a better understanding of the subject. The narrative could benefit from a more critical evaluation of the limitations and gaps in current research.

The review could be strengthened by a more detailed discussion on the methodological quality of the included studies. Assessing study design, sample size, and potential biases would provide a clearer picture of the reliability of the findings.

The review didn't follow PRISMA guidelines.

The objective was not clearly and adequate explained.

Incorporate a more critical analysis of the strengths and weaknesses of the current literature, including methodological assessments of the studies reviewed.

Add any figure to make understand the inosotol path.

You should mention comparative studies if any?

I suggest not to accept but asking major revision for improve and clarify by explaining the path, the previous studies, adding prisma, interpretation of the currents guidlines.

The similarity is about 35%. Should be pharaphrased and reducing the similarity.

Comments on the Quality of English Language

minor changes are needed.

Author Response

Answers to referees' questions

n.3

The reviewed manuscript is a narrative review discussing the potential roles of myoinositol and metformin in preventing gestational diabetes mellitus (GDM) in high-risk patients. The paper emphasizes the increasing prevalence of GDM due to rising global obesity rates and delayed pregnancies, particularly in industrialized countries. It explores the use of myoinositol and metformin as preventive measures, analyzing existing literature and studies to assess their efficacy.

While the review is comprehensive, it largely reiterates existing knowledge without offering new insights or critical analysis that could lead to a better understanding of the subject. The narrative could benefit from a more critical evaluation of the limitations and gaps in current research.

The review could be strengthened by a more detailed discussion on the methodological quality of the included studies. Assessing study design, sample size, and potential biases would provide a clearer picture of the reliability of the findings.

According to your suggestions we tried to report for each study the situations of weakness that could make the conclusions reached less reliable and all these were then reiterated in the conclusions where a multicenter study is hypothesized that overcomes all the weaknesses highlighted.

The review didn't follow PRISMA guidelines.

The PRISMA protocol is mandatory for systematic reviews but is not required for narrative reviews like this one.

The objective was not clearly and adequate explained.

According to your suggestion we have included a sentence at the beginning of the abstract that immediately clarifies the objective of the study which is to verify whether metformin or myoinositol have any efficacy in preventing the onset of GDM.

Incorporate a more critical analysis of the strengths and weaknesses of the current literature, including methodological assessments of the studies reviewed.

As previously mentioned we have tried to make explicit the limits of the studies presented from which we can then start again for a planned research that overcomes them. For example, in the RCT studies on myoinositol it has been clarified that they are almost all open-label and this can represent a bias.    Double-blind studies would have more statistical strength.

Add any figure to make understand the inosotol path.

According to your suggestion we have included a figure on the mechanism of action of myoinositol

You should mention comparative studies if any?

According to your suggestion we have mention some comparative studies (and reported in tables) although their limited significance compared to RCT trials is evident. For each drug we have reported in the text also some recent meta-analysis (but we have not included it in the tables).

Similarity

According to your suggestion we have rewritten some parts of the text in an attempt to limit the similarity, although when writing reviews, the due citations make the percentage of similarity naturally higher

Round 2

Reviewer 1 Report

Comments and Suggestions for Authors

I appreciate the work done by the authors

Reviewer 3 Report

Comments and Suggestions for Authors

Accept

Comments on the Quality of English Language

Accept